# Real world data on cervical cancer treatment patterns, healthcare access and resource utilization in the Brazilian public healthcare system

**Thabata Martins Ferreira Campuzano** [1] *, **Maria Amelia Carlos Souto Maior Borba** [1ᵒ],
**Paula de Mendonça Batista** [1ᵒ], **Michelle Nadalin** [1ᵒ], **Cicera Pimenta Marcelino** [1ᵒ], **Paula Cristina Pungartnik** [2‡], **Angélica Carreira dos Santos** [2‡], **Letícia Paula Leonart Garmatter** [1ᵒ], **Maria Aparecida do Carmo Rego** [1ᵒ], **Angélica Nogueira-Rodrigues** [3]

**1** MSD Brazil, São Paulo, Brazil, **2** IQVIA Brazil, São Paulo, Brazil, **3** Universidade Federal de Minas Gerais (UFMG), Minas Gerais, Brazil

ᵒ These authors contributed equally to this work.
‡ PCP and ACS also contributed equally to this work.
* thabata.martins@merck.com

## Abstract

The aim of the study is to evaluate the treatment patterns, time to start treatment, and healthcare resources utilization (HCRU) of cervical cancer (CC) patients within the Brazilian public health system (SUS). This is an observational retrospective study using SUS administrative database (DATASUS). Data from January-2014 to December-2020 was gathered from patients with the ICD-10 C53 codes. From 2014 to 2020, 206,861 women were included, among whom 90,073 (43.5%) had stage information. Of staged patients, 60.7% (54,719) had advanced disease (stages III and IV) and the most performed treatments were chemoradiotherapy (CRT) (41.6%), surgery + CRT (19.1%), radiotherapy (RT) only (16.8%) and chemotherapy (CT) only (13.3%). The proportion of patients submitted to CT in advanced stages was higher than in non-advanced stages (I and II), in contrast to RT, which was more frequent in stage I than stage IV. Median time to initiate treatment surpassed two months in approximately 30% of the cases, regardless of stage. Conization was the most performed surgical procedure. The hospitalization rate per patient per month for stage IV was twice as high as stage I (0.05 [95%CI 0.05–0.05] and 0.11 [0.11–0.11], respectively). The same trend was observed for outpatient visits (0.54 [95%CI 0.53–0.55] and 0.96 [0.93–0.98], respectively). This study demonstrated a high proportion of advanced CC at diagnosis in Brazil. The treatment pattern showed that chemoradiotherapy was the most frequent regimen overall and the use of chemotherapy and HCRU increased with staging. These results could provide information to improve public policies towards access to prevention, diagnosis, and treatment of CC in Brazil.

**Data Availability Statement:** All relevant data are within the manuscript and its Supporting Information files.

**Funding:** The author(s) received no specific funding for this work.

**Competing interests:** I have read the journal's policy and the authors of this manuscript have the following competing interests: TMFC, MACSM, PMB, MACR, MN, LPLG and CPM are employees of Merck Sharp Dohme Farmacêutica Ltda, Sao Paulo, Brazil, who may own stock and/or hold stock options in Merck & Co., Inc., Rahway, NJ, USA. ANR is a scientific medical consultant who was paid by MSD Brazil. PCP and ACS were employees of IQVIA Brazil, which was contracted by MSD Brazil to conduct the study.

# Background

Cervical Cancer (CC) is the fourth most common malignancy diagnosed in women worldwide, and the third most incident cancer among women in Brazil [1, 2]. In 2020, the number of new CC cases was 604,127 worldwide, and there were about 341,831 deaths [1]. In Brazil, the estimated incidence for each year of the 2023–2025 triennium is 17,010, with 6,627 deaths related to the disease reported in 2020 [3–5]. CC is one of the most common types of cancer in some states in the North region, with an estimated 20.48 cases per 100,000 women in 2023, followed by the Northeast and Middle East regions [3].

In 2020, the World Health Organization (WHO) adopted the 'Global strategy to accelerate the elimination of cervical cancer as a public health problem', which is based on three pillars: vaccination, screening, and treatment [6]. These pillars are prioritized to reduce morbidity, mortality, and costs related to cervical cancer [6]. In SUS (*Sistema Único de Saúde*), the Brazilian public health system, the quadrivalent Human Papillomavirus (HPV) vaccine has been available for girls aged 9 to 14 since 2014 [7, 8] and for boys aged 11 to 14 since 2017 [9]. However, vaccination coverage remains below the necessary threshold for CC elimination and is decreasing each year [10, 11].

For CC screening, the Brazilian guideline recommends the cytology exam by Papanicolaou smear (Pap smear) to detect precancerous abnormalities and prevent CC, for women aged 25 to 64 years old [9]. To date, the recommendation is to perform two consecutive annual exams and, in case of negative results, one every three years thereafter [12]. According to recommendation of the National Commission for the Incorporation of Technologies (CONITEC), HPV DNA screening should be available in the Brazilian public health system in 2024 [13], as per SEC-TICS/MS Ordinance No. 3 [14]. DNA-based testing for HPV has been shown to be more effective than commonly used screening methods [15]. In addition to being an effective technology for early detection and diagnosis, it has the advantage of increasing the interval between tests [16]. DNA screening is recommended every five years, bringing better adherence [16].

Despite national programs for CC prevention and screening, the disease is an economic burden for SUS, especially considering that most of Brazilian patients are still diagnosed with advanced stages [7, 17–19]. Disease stage is one of the major prognostic factors impacting survival. The CC 5-year overall survival rate is 59.4% for patients with early-stage disease and only 17.1% for advanced disease [20]. The recurrence rates may vary from 11% in early stages to 70% in advanced stages [21]. The preferred treatment to manage CC in early stages (I-II) is surgery with or without radiation therapy. For locally advanced/advanced disease (III-IV), the standard treatment is chemoradiotherapy (CCRT), which involves external beam radiation therapy combined with platinum-based chemotherapy, followed by brachytherapy [22–24]. While the gold standard treatment for advanced/metastatic stages is chemotherapy (CT) combined with immunotherapy, with or without targeted therapy [24–26], in Brazil, only CT is widely available in SUS, increasing the disparities in the healthcare of CC patients compared to the private setting.

Unlike some other countries, Brazil does not have specific guidelines for CC treatment within SUS, and there are discrepancies in the screening programs and tracking system and management at the public health system [12]. In addition to the delay in the inclusion of new oncology medications in public healthcare system, Brazil presents significant inequalities across the country, mainly related to social, economic, geographical and health access determinants, which have been associated with decreased efficiency in the detection of disease, treatment adherence and outcomes [27–30].

Creating strategies to reduce the CC burden in Brazil is fundamental to better understanding the CC scenario within SUS based on real-world evidence. Therefore, the aim of the

present study was to describe CC patients' journeys, including treatment patterns, time to start treatment, distance between the healthcare service and patient residence, demographic characteristics, and unprecedented results on the healthcare resource utilization by CC patients in Brazil using the administrative claim database DATASUS.

## Materials and methods

### Study setting

This was an observational retrospective study using the administrative claim databases DATA-SUS. DATASUS is the Informatics Department of SUS, the body responsible for collecting, processing, and disseminating healthcare data, providing public databases of inpatient and outpatient procedures. Despite the universal coverage of Brazilian public healthcare system, approximately 75% rely exclusively on it [31].

In this study, we used the following DATASUS databases: Inpatient Information System (SIH [*Sistema de Informações Hospitalares*]) and Outpatient Information System (SIA [*Sistema de Informações Ambulatoriais*]): SIA-AQ (a subdomain of SIA database specifically for CT procedures), SIA-AR (a subdomain of SIA database specifically for radiotherapy [RT] procedures) and SIA-BI (a subdomain of SIA database specifically for exams performed in SUS). Procedures are organized by the SUS Procedures, Medications and Orthoses, prosthetics, and special materials (OPM [*Órteses*, *próteses e materiais especiais*]), Table Management System (SIGTAP [*Sistema de Gerenciamento da Tabela de Procedimentos*, *Medicamentos e OPM do SUS]*]) within DATASUS and are identified by a 10-digit identification number for each procedure available at SUS. The disease stage information can be found in the SIA-AQ and SIA-AR databases.

### Population, information sources and criteria

Data from January 2014 to December 2020 was gathered for all patients with the International Classification of Diseases Tenth Revision for CC (ICD-10 code C53) [32]. The administrative claim data are presented as procedure codes from billing records and include demographic information, all procedures (inpatient and outpatient settings) performed by each patient, the number of procedures and other information. Procedures included diagnosis and treatment information, and only the records linked to ICD-10 code C53 were considered. DATASUS also provides information about the date of procedures and CC staging.

### Data extraction, inclusion and exclusion criteria

Probabilistic record linkage was performed using a multi-step approach with different combinations of patient-level information from Inpatient Information System—SIH) and Outpatient Information System—SIA), based on date of birth, city, and ZIP code. Before each step, data cleaning was performed to keep only high-quality claims for linkage. We estimated that about 5% of all patient records would be discharged from analysis due to low-quality information. This approach enabled an assessment of each patient's historical record and thus allowed the evaluation of their journey across the system.

After the probabilistic record linkage process, patients were considered eligible if $\geq 18$ years old with at least one claim of CC ICD-10 code C53 and at least six months of data during the study period. The index date was the first claim with ICD-10 C53 in the study period. Patients were excluded if they had inconsistent sex (male) or age (i.e., negative age or over 150 years old) and unknown treatment location/local of residence. Patients were also separated into those with and without staging information available. The stage was considered in the

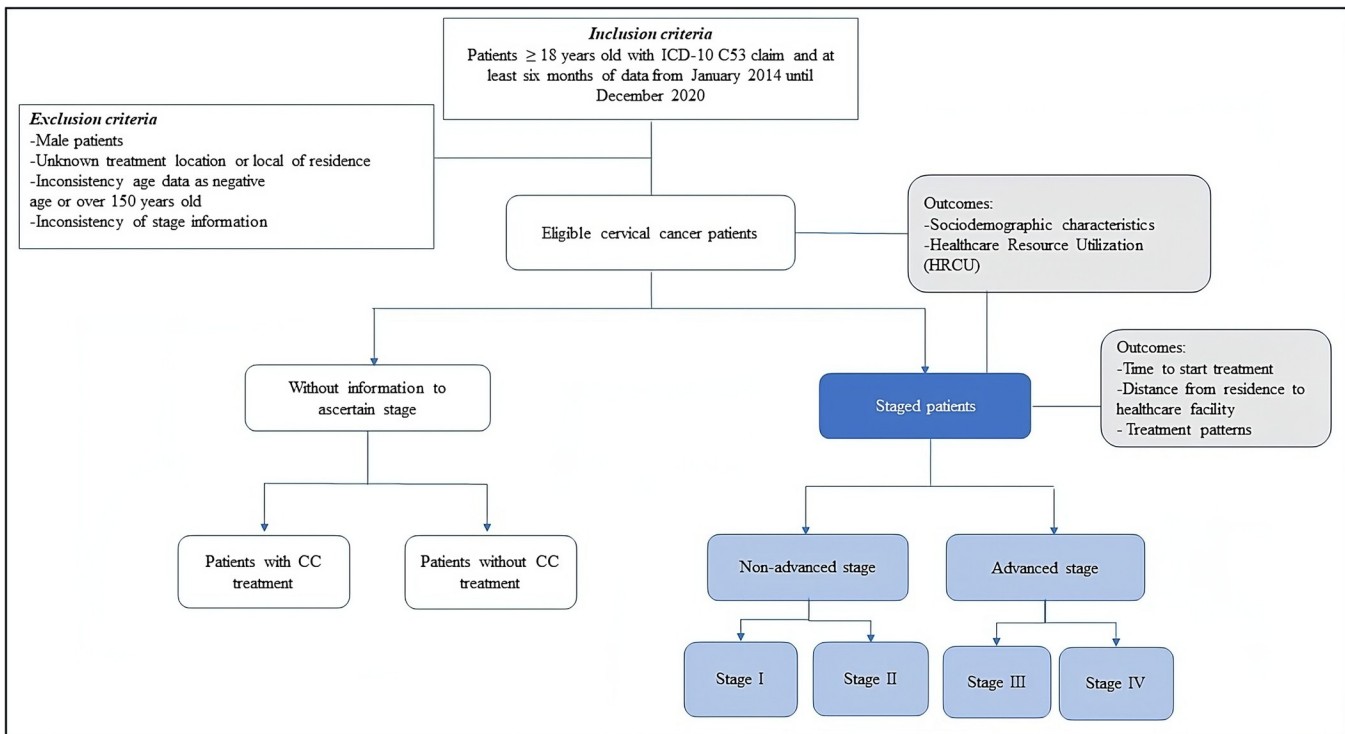

**Fig 1. Data process for CC cohort identification and staging classification of the CC patients.**

first ICD-10 reported in the database, and classified as non-advanced, for stages I-II, and advanced, for stages III-IV (Fig 1). Patients with inconsistent stage information (i.e., stage information different from I to IV) at the first CC claim were excluded (Fig 1).

The outcomes evaluated for all ICD-10 C53 patients were sociodemographic characteristics (age, ethnicity, residence state and region) and healthcare resource utilization (HCRU), measured as hospitalization rate, length of stay [LOS], outpatient visits, visits for chemotherapy, and visits for RT. These outcomes were also evaluated by stage of disease for patients who presented this information. Additionally, the staged patient's distance from residence (city level) to treatment institution (city level), time to start treatment (defined in this study as the time elapsed from the first claim of ICD-10 C53 to first treatment procedure [CT, RT or surgery– except conization/cone biopsy]), and treatment pattern were assessed.

## Data analysis

All outcomes were presented according to appropriate statistics: continuous variables (quantitative ones) were summarized by mean, standard deviation (SD), minimum, maximum, median, and interquartile range (IQR) (25th and 75th percentiles), while categorical variables were described by simple and cross-contingency tabulation, with absolute frequencies and percentages with 95% confidence interval (CI). The HCRU per patient was summarized as the mean (SD) and median (IQR) number of procedures per each patient; and HCRU per-patient per-month (PPPM) was calculated as the median (95% CI) number of procedures divided by each patient's follow-up time in months ($PPPM = \frac{N\ procedures}{FUP}$). Thus, the longer the follow-up, the lower the PPPM, even if there are many procedures claims.

Treatment patterns were evaluated descriptively. Frequencies, percentages, and 95% CIs were presented based on the treatment sequences and most frequently reported treatments for each line of treatment (LOT) using a Sankey diagram.

## Results

After applying the inclusion and exclusion criteria, 206,861 patients were identified in DATA-SUS. Of these, 90,073 patients (43.5%) presented staging information (Fig 2).

The median age at index was 49.5 years (interquartile range [IQR] 38–61). The stage I (53.09 years [IQR 41.18–64.14]) and stage IV (54.19 years [IQR 43.53–64.28]) patients had the highest median age at index (Table 1).

Among patients with disease stage information (n = 90,073), 60.7% presented advanced disease (n = 54,719) at first CC claim in the evaluated period (Fig 2). The South region had the highest proportion of advanced disease, 36.3% for stage III and 28.1% for stage IV. The Northeast had the highest proportion of stage III CC (45.3%) compared to the other Brazilian regions. In this region, the proportion of stage IV was 16.9% (Fig 3A). The North region had the lowest proportion of advanced disease, 33.8% for stage III and 16.6% for stage IV. CC stage III predominated across all Brazilian regions, apart from the North region, in which stage II stood out.

### Health care resource utilization–inpatient

Among the 206,861 patients, 32.4% (n = 66,995) had at least one record of hospitalization (median 1.0, IQR 1.0–2.0). The median hospitalization per patient per month was 0.14 (95%CI 0.14–0.14). The median LOS per hospitalization was 4.0 days (IQR 2.0–7.0) (Table 2).

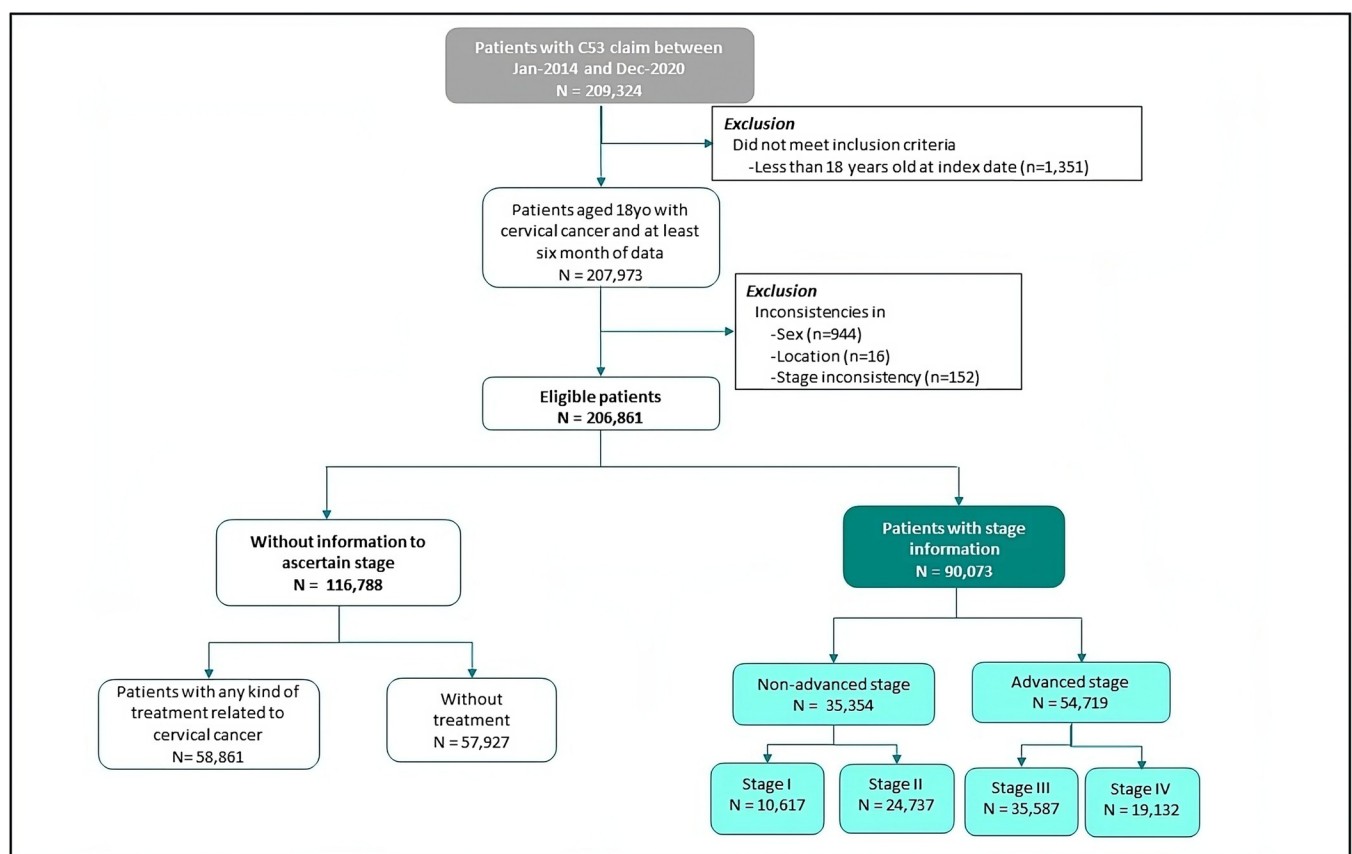

**Fig 2. Flowchart of the trial patient cohort selection process.**

**Table 1. Description of sociodemographic characteristics of included patients.**

|  | All patients* | **Non-advanced | | **Advanced | |
|---|---|---|---|---|---|
|  |  | Stage I | Stage II | Stage III | Stage IV |
| **N (%)** | 206,861 | 10,617 (5.1) | 24,737 (12) | 35,587 (17.2) | 19,132 (9.2) |
| **Age** (years) |  |  |  |  |  |
| Mean (SD) | 49.91 (15.46) | 53.22 (14.65) | 52.56 (14.64) | 52.71 (14.4) | 54.11 (13.99) |
| Median (IQR) | 49.45 (37.92–60.98) | 53.09 (41.18–64.14) | 51.77 (40.86–63.13) | 52.22 (41.35–62.93) | 54.19 (43.53–64.28) |
| **Age** n (%) |  |  |  |  |  |
| 18 to 30 years | 24,298 (11.7) | 544 (5.1) | 1,332 (5.4) | 1,881 (5.3) | 833 (4.4) |
| 31 to 50 years | 86,702 (41.9) | 4326 (40.7) | 10,596 (42.8) | 14,865 (41.8) | 7,189 (37.6) |
| 51 to 70 years | 75,556 (36.5) | 4424 (41.7) | 9,884 (40.0) | 14,685 (41.3) | 8,736 (45.7) |
| > 70 years | 20,305 (9.8) | 1323 (12.5) | 2,925 (11.8) | 4,156 (11.7) | 2,374 (12.4) |
| **Ethnicity** n (%) |  |  |  |  |  |
| White | 78,804 (38.1) | 4,841 (45.6) | 8,881 (35.9) | 13,012 (36.6) | 8,687 (45.4) |
| Black | 11,133 (5.4) | 569 (5.4) | 1,276 (5.2) | 2,025 (5.7) | 1,074 (5.6) |
| Mixed | 75,855 (36.7) | 4,283 (40.3) | 12,112 (49.0) | 17,669 (49.7) | 7,955 (41.6) |
| Indigenous | 341 (0.2) | 10 (0.1) | 22 (0.1) | 65 (0.2) | 25 (0.1) |
| Asian | 12,449 (6.0) | 605 (5.7) | 1,545 (6.2) | 1,903 (5.3) | 869 (4.5) |
| *Missing information* | *28,279 (13.7)* | *309 (2.9)* | *901 (3.6)* | *913 (2.6)* | *522 (2.7)* |

*Includes all patients with stage 1–4, and those non-staged.** A total of 90,073 patients had stage information available.

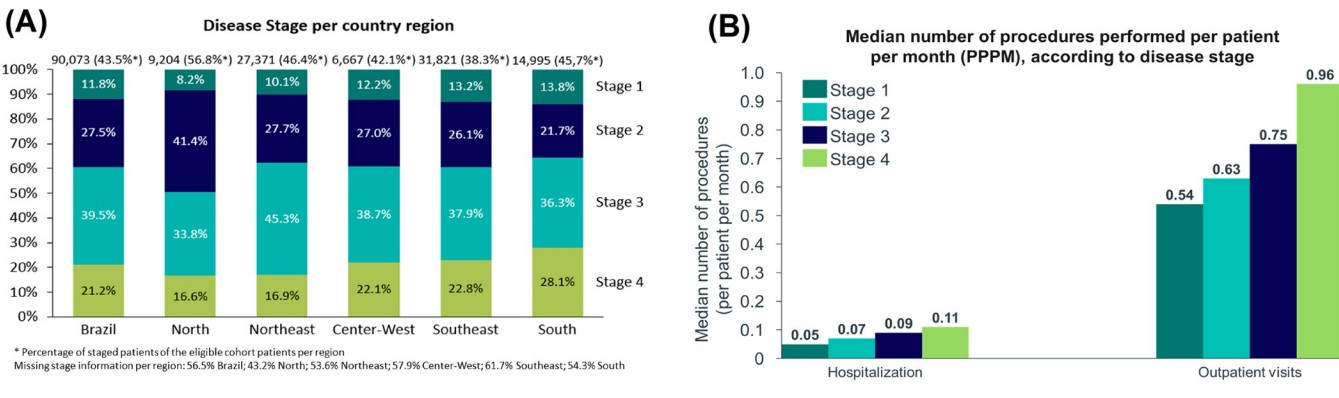

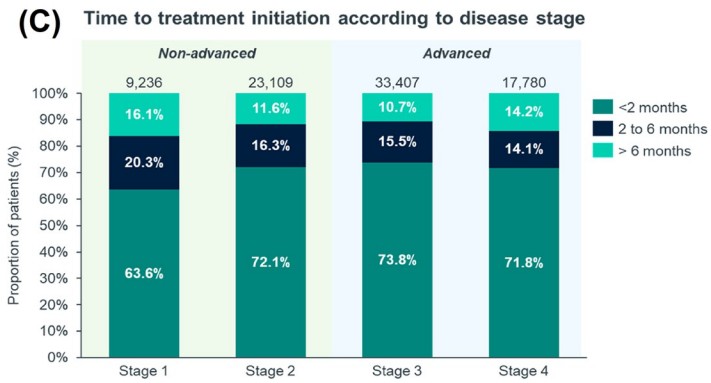

**Fig 3. Brazilian region distribution, treatment access and healthcare resource utilization of cervical cancer patients, per disease stage.** (A) Disease stage per country region in % of patients at first claim from staged cohort; (B) Healthcare resource utilization per patient per month of cervical treatment in SUS per disease stage; (C) Time to start treatment in the SUS per disease stage.

**Table 2. Healthcare resource utilization according to disease stage and type of procedure.**

| | All patients* | **Non-advanced | | **Advanced | |
|---|---|---|---|---|---|
| | | Stage I | Stage II | Stage III | Stage IV |
| **N (%)** | 206,861 (100) | 10,617 (5.1) | 24,737 (12) | 35,587 (17.2) | 19,132 (9.2) |
| **Inpatient setting** | | | | | |
| N (%) | 66,995 (32.4) | 4,115 (38.8) | 8,718 (35.2) | 14,779 (41.5) | 8,693 (45.4) |
| **Hospitalization** | | | | | |
| Mean (SD) per patient | 1.87 (1.83) | 1.94 (1.92) | 2.19 (2.33) | 2.34 (2.26) | 2.37 (2.03) |
| Median (IQR) per patient | 1.0 (1.0–2.0) | 1.0 (1.0–2.0) | 1.0 (1.0–3.0) | 2.0 (1.0–3.0) | 2.0 (1.0–3.0) |
| Median PPPM [CI] | 0.14 [0.14–0.14] | 0.05 [0.05–0.05] | 0.07 [0.07–0.07] | 0.09 [0.09–0.10] | 0.11 [0.11–0.11] |
| LOS per hospitalization (days) | | | | | |
| Mean (SD) | 6.01 (7.47) | | | | |
| Median (IQR) | 4.0 (2.0–7.0) | 5.84 (7.31) | 6.07 (7.4) | 6.46 (7.82) | 6.56 (7.71) |
| | | 3.0 (2.0–7.0) | 4.0 (2.0–7.0) | 4.0 (2.0–8.0) | 4.0 (2.0–8.0) |
| **Outpatient setting** | | | | | |
| Outpatient visit | | | | | |
| N (%) | 183,366 (88.6) | 10,009 (94.3) | 23,414 (94.7) | 33,816 (95) | 18,220 (95.2) |
| Mean (SD) per patient | 19.97 (27.96) | 24.63 (29.74) | 23.85 (31.34) | 23.9 (30.66) | 27.41 (36.41) |
| Median (IQR) per patient | 12.0 (5.0–23.0) | 16.0 (7.0–31.0) | 14.0 (7.0–28.0) | 15.0 (7.0–28.0) | 16.0 (7.0–33.0) |
| Median PPPM [CI] | 0.54 [0.54–0.54] | 0.54 [0.53–0.55] | 0.63 [0.62–0.64] | 0.75 [0.75–0.76] | 0.96 [0.93–0.98] |
| Outpatient Follow-Up (months) | | | | | |
| Mean (SD) per patient | 33.64 (24.38) | 38.41 (23.9) | 33.52 (23.31) | 29.93 (22.88) | 28.44 (23.09) |
| Median (IQR) per patient | 31.9 (10.98–53.93) | 38.92 (17.9–58.85) | 30.92 (12.98–51.93) | 24.98 (9.97–47.9) | 21.97 (8.95–45.9) |
| **Visits for Chemotherapy** | | | | | |
| N (%) | 71,687 (34.7) | 4,996 (47.1) | 19,038 (77) | 29,365 (82.5) | 16,271 (85) |
| Mean (SD) per patient | 6.13 (8.39) | 7.9 (12.0) | 5.6 (8.76) | 5.82 (7.9) | 6.8 (7.38) |
| Median (IQR) per patient | 3.0 (3.0–6.0) | 3.0 (3.0–7.0) | 3.0 (3.0–5.0) | 3.0 (3.0–6.0) | 5.0 (3.0–8.0) |
| Median PPPM [CI] | 1.48 [1.48–1.48] | 1.48 [1.48–1.48] | 1.50 [1.50–1.50] | 1.48 [1.48–1.48] | 1.20 [1.20–1.20] |
| Chemotherapy Follow-Up (months) | | | | | |
| Mean (SD) per patient | 7.43 (11.93) | 9.12 (14.56) | 6.82 (12.24) | 7.27 (11.75) | 8.05 (11.04) |
| Median (IQR) per patient | 2.0 (2.0–6.98) | 2.03 (2.0–8.03) | 2.0 (1.97–5.02) | 2.0 (2.0–6.95) | 4.0 (2.0–9.02) |
| **Visits for Radiotherapy** | | | | | |
| N (%) | 78,024 (37.7) | 9,533 (89.8) | 22,078 (89.3) | 30,717 (86.3) | 12,743 (66.6) |
| Mean (SD) per patient | 3.19 (1.85) | 3.34 (2.25) | 3.35 (1.87) | 3.23 (1.75) | 2.78 (1.66) |
| Median (IQR) per patient | 3.0 (2.0–4.0) | 3.0 (2.0–4.0) | 3.0 (2.0–4.0) | 3.0 (2.0–4.0) | 2.0 (2.0–4.0) |
| Median PPPM [CI] | 1.50 [1.50–1.50] | 1.50 [1.50–1.50] | 1.50 [1.50–1.50] | 1.50 [1.50–1.50] | 1.50 [1.49–1.50] |
| Radiotherapy Follow-Up (months) | | | | | |
| Mean (SD) per patient | 3.0 (5.01) | 2.62 (4.48) | 3.16 (4.94) | 3.07 (4.99) | 2.94 (5.69) |
| Median (IQR) per patient | 2.0 (1.0–3.02) | 2.0 (1.0–2.92) | 2.0 (1.02–3.02) | 2.0 (1.02–3.02) | 1.93 (1.0–2.03) |

LOS: length of stay; IQR: interquartile range; CI: confidence interval; SD: standard deviation; PPPM: per patient per month.

* Includes all patients with stage 1–4, and those non-staged.

** A total of 90,073 patients had stage information available.

The proportion of patients with at least one hospitalization record increased from stage I (38.8%) to stage IV (45.4%). The median number of hospitalizations per patient showed an increasing trend according to disease stage (from 1.0 [IQR 1.0–2.0] in stage I to 2.0 [IQR 1.0–3.0] in stage IV). Stage IV CC patients presented a median hospitalization per patient per month twice as high as stage I (0.05 [95% CI 0.05–0.05] and 0.11 [0.11–0.11], respectively)

(Fig 3B and Table 2). CC stage I patients had the shortest LOS during hospitalization (3.0 days [IQR 2.0–7.0]), while CC stages II to IV has LOS was 4 days (IQR 2.0–8.0) (Table 2).

## Health care resource utilization–outpatient

For the outpatient visits, 88.6% (n = 183,366) of patients had at least one record (not including visits for CT or RT procedures), with a median number of outpatients visits per patient of 12.0 (IQR 5.0–23.0) (Table 2). The median hospitalization per patient per month, was 0.54 (95% CI 0.54–0.54).

The proportion of patients with a record of outpatient visits was similar across all disease stages (Table 2). However, stage IV CC had twice as many outpatient visits per patient per month as compared to stage I (0.54 [95%CI 0.53–0.55] and 0.96 [0.93–0.98], respectively) (Fig 3B and Table 2).

## Health care resource utilization—visits for chemotherapy

Around 34.7% (n = 71,687) of all CC patients (n = 206,861) had at least one record of visit for chemotherapy. The median number of visits per patient was 3.0 (IQR 3.0–6.0) and median number of visits was 1.48 (95% CI: 1.48–1.48) per patient per month (Table 2).

The proportion of patients with record of visits for chemotherapy increased from stage I (47.1%) to stage IV (85.0%). Stage IV patients had a higher median number of visits for chemotherapy per patient (5.0 [IQR 3.0–8.0]) and the smallest number of visits for chemotherapy per patient per month (1.20 [95%CI 1.20–1.20]) (Table 2).

## Health care resource utilization—visits for radiotherapy

Of the 206,861 patients, 37.7% (n = 78,024) had at least one record of visits for RT. The median number of visits per patient was 3.0 (IQR 2.0–4.0) and the median number of visits for RT was 1.50 (95% CI: 1.50–1.50) per patient per month, for all patients (Table 2).

The proportion of patients with record of visits for RT decreased from stage I (89.8%) to stage IV (66.6%). Stage IV patients had a smaller median number of visits for RT per patient (2.0 [IQR 2.0–4.0]) (Table 2).

## Time to treatment initiation

The median time to initiate treatment from the index date was similar between non-advanced (1.02 months [IQR 1.02–3.02]) and advanced (1.02 months [IQR 0.98–2.95]) disease, and across all Brazilian regions (S1 Table). Most patients with non-advanced disease (69.7%) started treatment within 2 months after first claim of CC, which is the maximum time for SUS to provide the treatment, according to Brazilian legislation [25]. The most significant delay in treatment initiation was observed for stage I CC, in which 36.4% of patients had to wait more than 2 months to receive treatment (Fig 3C).

## Distance from residence to healthcare facility

Most medical visits to healthcare facilities (72.6%) were performed up to 50 km (about 31.07 mi) from the patient's home (S2 Table).

The proportion of visits that took more than 50 km (about 31 mi) from the patient's home to the healthcare facility was higher among those with advanced disease. Patients with stage III and IV traveled a greater median distance of 7.96 km [IQR 0.0–89.16] and 4.88 km [IQR 0.0–75.16], respectively, compared to the non-advanced stages (S2 Table).

## Treatment patterns

All the staged patients had claims for treatment procedures (surgery, CT and/or RT). The most performed treatments were CRT (41.6%), surgery + CRT (19.1%), RT only (16.8%) and CT only (13.3%) (Fig 4A). Conization was the most performed surgical procedure and more than 80% underwent the procedure only once. The second most frequent surgery was hysterectomy with pelvic exenteration, higher for stage I (42.1%). About 90% of stage I patients versus 67% of stage IV patients were submitted to RT (Fig 4B). On the other hand, the proportion of patients submitted to CT increased with staging, from 47.1% in stage I to 85.0% in stage IV (Fig 4B). Pelvic lymphadenectomy was the most common approach in stage I (87.5%) while the retroperitoneal approach was more frequent among stages II (46.7%), III (48.8%) and IV (37.1%) (S3 Table).

Regarding CT treatment, as observed in the Sankey diagram, switching between therapies was more frequent in advanced stages (Fig 5A and 5B), while in stages I and II, most patients had only one line of treatment. Stage I and II (non-advanced) patients received mostly cisplatin (68.3%) and paclitaxel (12.9%) as first-line therapy. The most frequent subsequent LOTs in recurrent/progressive cases were paclitaxel (35.5%), cisplatin + paclitaxel (9.3%) and other therapies (32.8%) as LOT2, and paclitaxel (17.0%), gemcitabine (19.5%) and other therapies as LOT3 (41.4%) (Fig 5A and 5B).

For stage III patients (Fig 6A), the most frequent LOT1 therapies were cisplatin (63.1%) and paclitaxel (15.7%). The regimens with cisplatin+gemcitabine (2.7%) and cisplatin+paclitaxel (2.3%) were also more frequent as LOT1 in stage III patients compared to the previous stages. The most received LOT2 therapies for stage III was paclitaxel (39.0%), while in LOT3 less concordance was observed, and the most received therapy were paclitaxel (19.3%), and gemcitabine (17.2%) (Fig 6A). Cisplatin (33.5%) and paclitaxel (35.9%) were the most frequent therapies for stage IV patients as LOT1, but compared to the less advanced stages, cisplatin was much less representative (Fig 6B). The proportion of patients receiving cisplatin+paclitaxel (8.6%), cisplatin+gemcitabine (2.7%) and gemcitabine (2.8%) was higher in stage IV LOT1 compared to non-advanced stages. In addition to paclitaxel and other therapies, gemcitabine was one of the most frequent regimens for stage IV patients in LOT2 (15.1%) and LOT3 (17.1%) (Fig 6B).

## Discussion

This is a real-world national descriptive study of CC patients' journeys in the Brazilian public healthcare system. So far, this is the largest evaluation of HCRU for CC treatment in Brazil, using 7 years of data, between 2014 and 2020.

Although there have been improvements in the prevention of CC in Brazil with the expansion of the screening program and the availability of the vaccine, diagnosis and access to treatment remain determinants of CC burden [28]. Similar to the findings of this study, other Brazilian studies reported a mean age of 50 years in CC patients, and most patients with stage information available are already diagnosed with advanced disease at first contact with the health system, reflecting the delay in CC diagnosis [17–19, 28]. Furthermore, this younger profile of CC patients in Brazil increases the socio-economic disease burden, as women retire after 65 years old, i.e., CC can lead to an early exit from the job market by approximately 15 years due to absenteeism [33].

Personal and structural barriers, like lack of education, income inequality, fear, distrust, cost, and limited access to health care are well-stablished determinants that hamper CC screening in low- and middle-income countries such as Brazil [27–29]. In addition, geographical accessibility has been described as one of the most crucial factors to health access which is

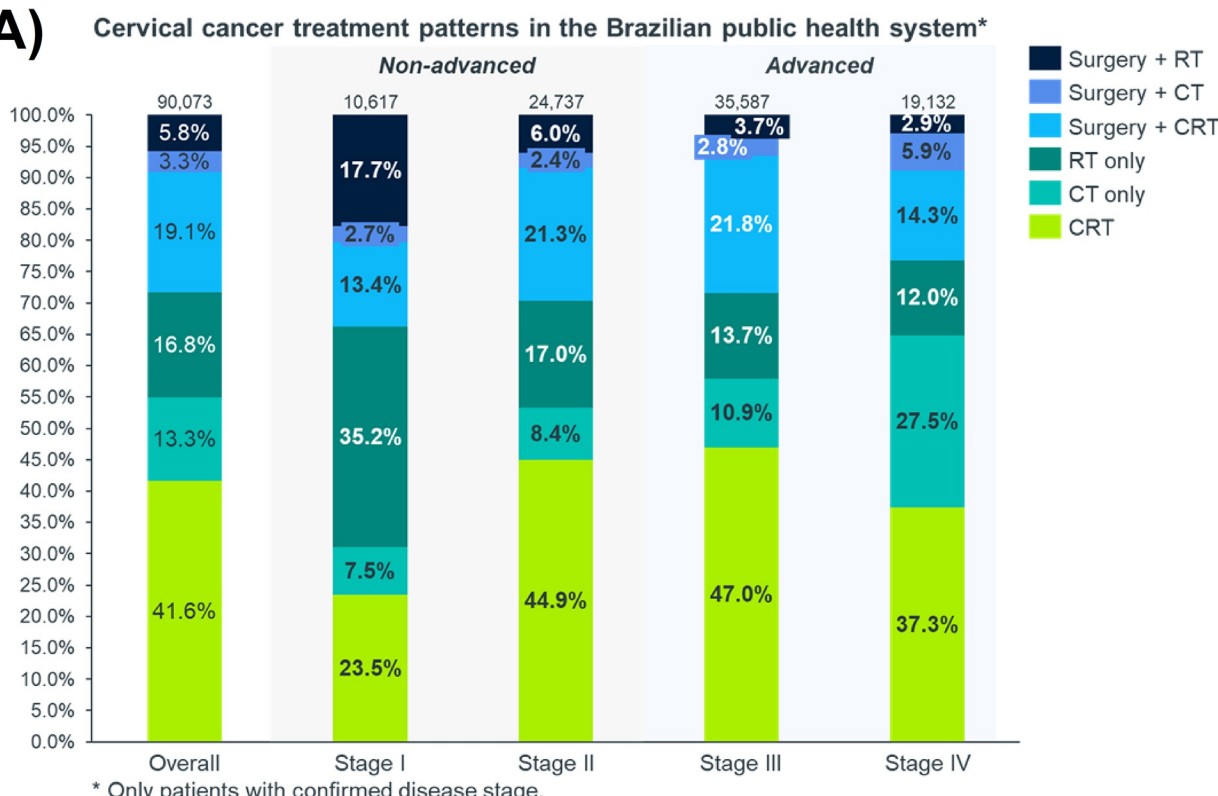

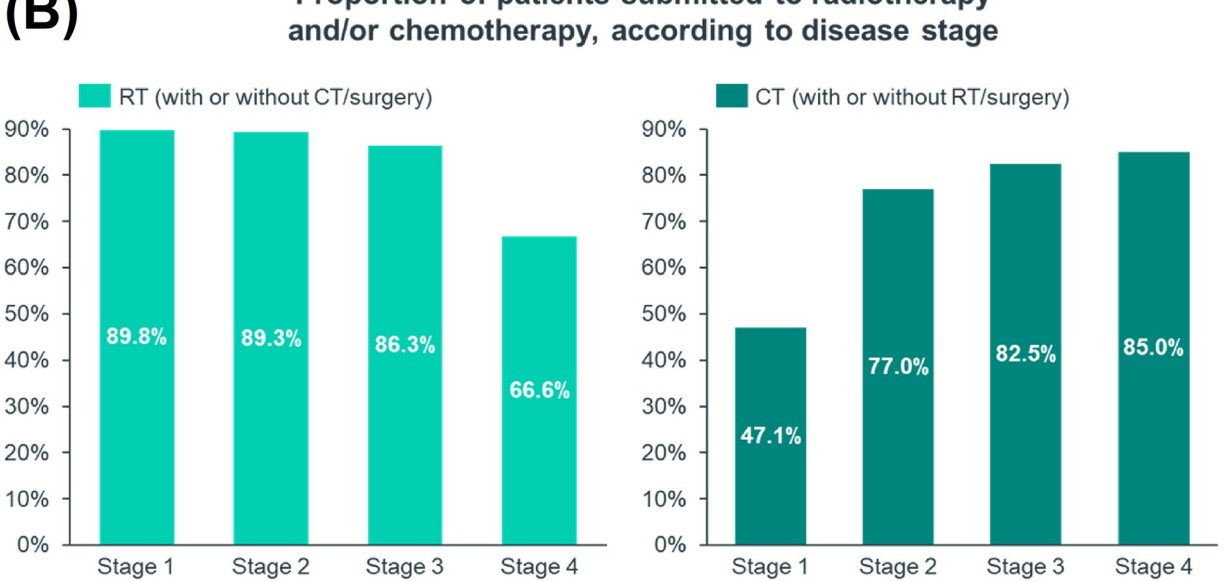

**Fig 4. Cervical cancer treatment patterns according to disease stage.** (A) Cervical cancer treatment according to disease stage; (B) Proportion of patients submitted to radiotherapy and/or chemotherapy per disease stage.

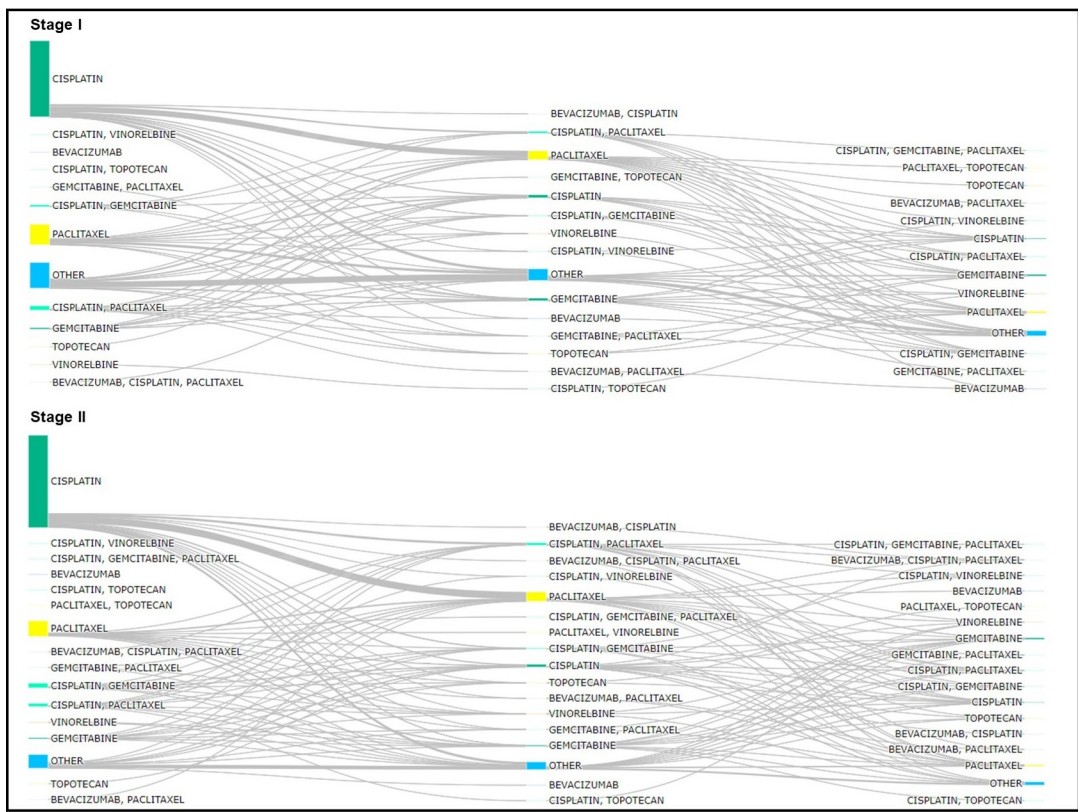

**Fig 5. Sankey diagram of treatment patterns for cervical cancer stage I and II (non-advanced).** (A) Sankey diagram of treatment patterns for cervical cancer stage I; (B) Sankey diagram of treatment patterns for cervical cancer stage II.

associated with time to disease diagnosis, initiating treatment, and treatment adherence [34]. Several difficulties have been reported among cancer patients who had to commute for treatment, such as fatigue, long waits to return home, lack of proper feeding, lack of financial resources to travel, and a continuous interruption of routine activities [35]. In this study, most medical visits were performed up to 50 km (about 31.07 mi) from the patient's residence, and patients with stage III and IV traveled a greater distance than the non-advanced ones. These results are in line with other previous studies published in the literature in the Brazilian setting [18, 36], and in other low- and middle-income countries, such as Malaysia, Colombia, Trinidad and Tobago [37–39].

In Brazil, access to cancer treatment is widely influenced by the concentration of services in large urban centers, and consequently by the large distances travelled by patients [36]. A study published in 2022 about geographic accessibility to cancer treatment in Brazil identified marked differences among the Brazilian regions, where patients living in the North and Midwest regions had to travel greater distances compared to other Brazilian regions [36]. The higher concentration of patients in wealthier Brazilian regions reinforces the relationship between greater access to healthcare and earlier disease diagnosis, which can result in better outcomes [39]. These areas are characterized by higher per-capita income, elevated educational levels, concentration of health services, access to reference medical units, and higher per-capita health resources in both private and public sectors [39]. In contrast, it was unexpected to find a higher proportion of non-advanced patients in the North region, as this region was reported to have the highest mortality of CC over time, consistently above the national

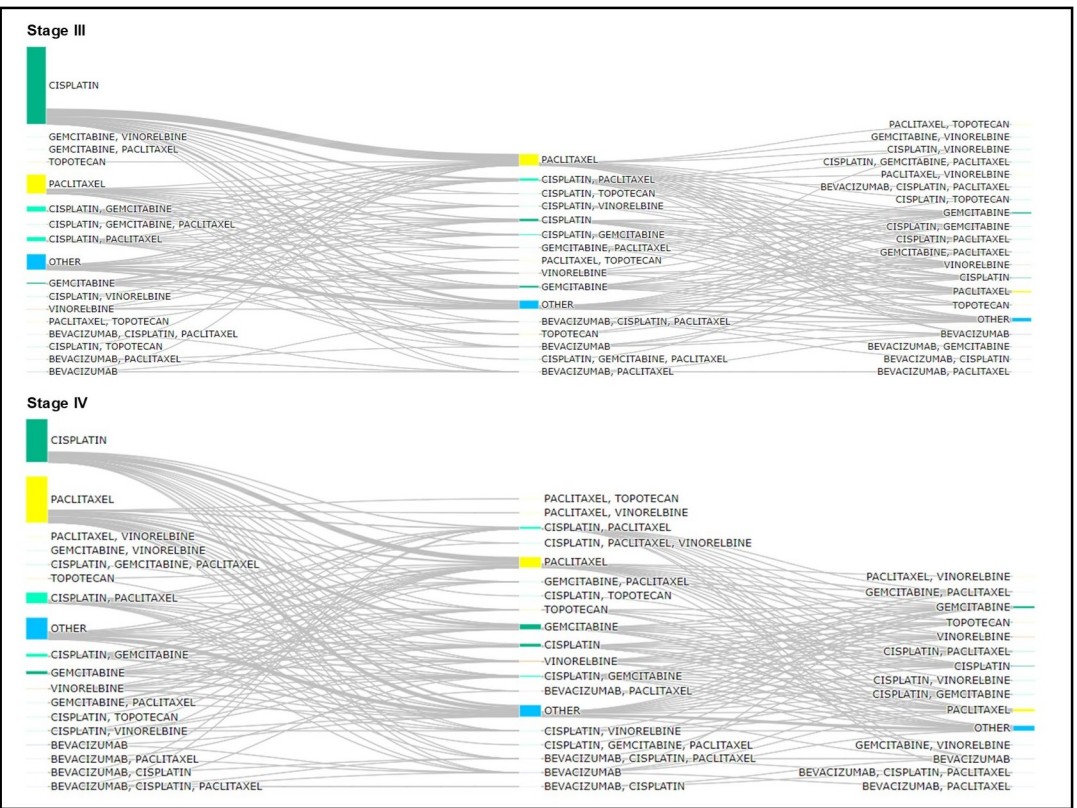

**Fig 6. Sankey diagram of treatment patterns for cervical cancer stage III and IV (advanced).** (A) Sankey diagram of treatment patterns for cervical cancer stage III; (B) Sankey diagram of treatment patterns for cervical cancer stage IV.

and other regional trends [40]. However, it has been reported that efforts have been made in this region that increased the population's adherence to CC screening programs. For instance, a study conducted in Boa Vista, Roraima (North region) demonstrated a screening coverage rate of 85.7%. On the other hand, early diagnosis does not necessarily mean access to treatment, which is necessary to achieve mortality reduction [41].

The delay between the diagnosis and the beginning of treatment also reinforces the importance of understanding the consequences of geographic accessibility. Even though it has been mandatory since 2012 for all cancer patients to start treatment within 60 days from the confirmed histological diagnosis [42], this analysis shows that it is not yet a reality for all patients diagnosed with CC. Corroborating this data, another Brazilian study reported a longer delay in treatment initiation for non-advanced CC stages [18]. This delay may result in an increase in cancer-related death risk and in higher HCRU, as costs increase for treating more advanced diseases [43–47], and may be related to structural (patterns of healthcare use and organization of primary and secondary care services within SUS) [36, 45] and nonstructural (related to the patient and physician choices) barriers [48].

Regarding treatment patterns, as expected, the proportion of patients undergoing surgery and RT decreases while CT increases as the stage of disease progresses. In the early stages (I-II), CC can be treated by surgery or radiation therapy, while for advanced stages (III-IV), chemotherapy or chemoradiation are usually more recommended [25, 49, 50]. Related to CT regimens, as expected, cisplatin—recommended as a first-line option for the treatment of non-advanced CC [51]—was the main prescribed CT for initial disease and was used in association

with RT in more than 88% of the patients. In clinical practice, patients with advanced disease treated in SUS generally receive paclitaxel regimens associated to cisplatin and other drugs, along with RT [25, 51–53], as observed in this study. It is important to highlight that during this study period (2014–2020), adjuvant chemotherapy was still used in CC treatment. In 2021, however, new evidence showed that adjuvant chemotherapy given after standard cisplatin-based chemoradiation for women with locally advanced CC had no effect on overall survival or progression-free survival, changing the management of the disease [54]. As some therapies are not available in SUS, bevacizumab appears occasionaly in CC treatment.

The HCRU results of this study corroborate with evidence from national and international literature, showing an increase in HCRU with the advancement of the disease [43, 44, 55, 56]. Studies in high-income countries have shown that treatment for cancer patients who have been diagnosed earlier are two to four times less expensive compared to treating people diagnosed with cancer at more advanced stages [57]. In 2022, a study about the economic burden among 1,229 CC patients in the US reported that HCRU for patients at LOT1 was smaller compared to patients at LOT3. Patients at LOT1 had a mean number of inpatient visits of 0.14 per patient per month while LOT3.

Patients had 0.22 PPPM inpatient visits. This was also observed for emergency visits and inpatient length of stay. The mean PPPM total costs incurred by patients with CC during each LOT were US$15,892 in LOT1, US$20,193 in LOT2 and US$16,576 in LOT3+ [54]. Although this study did not compare HCRU among different LOTs, an increase in HCRU was also identified with the advancement of the disease. Hospitalization, length of stay and outpatient visits also had this same increasing trend with disease advancement. Furthermore, in addition to the direct costs of managing CC, the economic burden related to productivity loss, disability and infertility of young women diagnosed with CC must also be considered [58, 59]. These indirect costs are not routinely evaluated but cannot be disregarded given their great social, economic, and epidemiological relevance.

Although national CC prevention policies were put in place in Brazil and no recent country-level evaluation of its effectiveness was performed, some city and regional-level evaluations already show promising results on increasing access to screening and reducing incidence and mortality [60, 61]. The importance of policies that tackle CC personal and structural barriers are highlighted by strengthened by the WHO call to eliminate CC as a public health problem by 2030 [62]. Despite the HPV quadrivalent vaccine has been available in SUS since 2014, its immediate effects cannot yet be observed. While the vaccine targets younger age groups, the screening program usually starts at 25 years old, and the disease mostly appears in advancing age, over 50 years old [63–65]. Therefore, there is a need for long-term surveillance of herd immunity effects, as well as improvements in treatment standards and outcomes in the country.

The results of this study should be interpreted in the context of its limitations. The major limitation of retrospective data is the probability of incomplete data and the variable quality of non-mandatory data, which is intrinsic to retrospective database studies. Additionally, the databases used in this study are administrative and reimbursement-oriented and therefore lack detailed, granular clinical data [66]. This underscores the need for more complete and efficient methods for collecting and managing this type of data, given its significance for understanding population characteristics. It is also important to mention that the healthcare resource utilization related to CC does not represent the entire Brazilian population, as approximately 75% rely exclusively on it [31].

More than half of women's records identified through inclusion criteria (56%) did not present staging information, which can somehow impact the results. To fully understand the scenario of cancer in Brazil and minimize bias, it is important that staging information be

consistently reported in DATASUS when a diagnosis of cancer is made or is under investigation. DATASUS only stratifies the stages from I to IV, not considering stage 0 or subcategories between stages. It is important to highlight that the analysis of conization (provided by SIH database) according to stage (provided by SIA-AQ and SIA-AR databases) might be influenced by the temporality of data related to the linkage process, since each information is provided from different databases. Patients may undergo conization procedures in an initial stage, however, they may be included in SIA-AQ and SIA-AR databases after an advanced stage diagnosis. Thus, the information of conization per stages implies that a patient, reported during study period as an advanced stage, underwent conization at another moment of the study. Additionally, the analysis of the time lag between the diagnosis and the beginning of treatment might be overestimated, as time to conization was not considered in the analysis, and patients submitted only to conization procedures might have a smaller time lag from diagnosis to conization.

## Conclusions

The economic and social burden of CC in Brazil is significant. This study reiterates the urgent call for assertive public policies to mitigate disparities in CC screening and prevention, such as increasing the coverage of HPV immunization in the target population (girls aged 9 to 14, and boys aged 11 to 14) and more effective screening of women aged 25 to 64 years. Early detection could impact the overall HCRU of CC, reducing costs and allowing for optimal allocation of resources to provide Brazilian patients with the best cancer care. Thus, the results presented in this study can provide valuable information for key stakeholders to make more informed health decisions, improving public policies and access to equitable prevention, diagnosis, and treatment of CC in Brazil. Additionally, this real-world analysis highlights the need to improve the quality of DATASUS to consistently capture the CC disease staging. Such improvements are essential for better understanding and resource management, towards achieving the WHO's CC elimination goals.

## Supporting information

**S1 Table. Time to treatment initiation according to disease stage and country region, from staged cohort.** *From first claim of C53 to treat procedure (surgery/CT/RT), except conization.
(DOCX)

**S2 Table. Distance from residence to healthcare facility, per visit.** *(Euclidean) distance is calculated based on the city of residence and of the institution (latitude and longitude). 1 Number of visits to the health center categorized by distance. The percentage reflects the proportion of the total number of visits.
(DOCX)

**S3 Table. Cervical cancer treatment according to disease stage.** *Patients may undergo multiple surgical procedures. **Surgery: hysterectomy or trachelectomy (if no hysterectomy procedure). *** A total of 90,073 patients had stage information available.
(DOCX)

## Author Contributions

**Conceptualization:** Thabata Martins Ferreira Campuzano, Maria Amelia Carlos Souto Maior Borba, Paula de Mendonça Batista, Michelle Nadalin, Maria Aparecida do Carmo Rego, Angélica Nogueira-Rodrigues.

**Formal analysis:** Thabata Martins Ferreira Campuzano, Maria Amelia Carlos Souto Maior Borba, Paula Cristina Pungartnik, Angélica Carreira dos Santos, Letícia Paula Leonart Garmatter, Angélica Nogueira-Rodrigues.

**Funding acquisition:** Paula de Mendonça Batista, Maria Aparecida do Carmo Rego.

**Investigation:** Angélica Nogueira-Rodrigues.

**Methodology:** Thabata Martins Ferreira Campuzano, Maria Amelia Carlos Souto Maior Borba, Letícia Paula Leonart Garmatter.

**Project administration:** Paula de Mendonça Batista, Cicera Pimenta Marcelino, Angélica Carreira dos Santos.

**Supervision:** Thabata Martins Ferreira Campuzano, Maria Amelia Carlos Souto Maior Borba, Paula de Mendonça Batista, Cicera Pimenta Marcelino, Paula Cristina Pungartnik, Angélica Carreira dos Santos, Letícia Paula Leonart Garmatter, Maria Aparecida do Carmo Rego.

**Validation:** Thabata Martins Ferreira Campuzano, Paula Cristina Pungartnik, Letícia Paula Leonart Garmatter.

**Writing – original draft:** Thabata Martins Ferreira Campuzano, Paula Cristina Pungartnik, Angélica Carreira dos Santos.

**Writing – review & editing:** Thabata Martins Ferreira Campuzano, Maria Amelia Carlos Souto Maior Borba, Paula de Mendonça Batista, Michelle Nadalin, Cicera Pimenta Marcelino, Paula Cristina Pungartnik, Angélica Carreira dos Santos, Letícia Paula Leonart Garmatter, Maria Aparecida do Carmo Rego, Angélica Nogueira-Rodrigues.

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
