## [Decision Letter · Decision Letter 0]

10 Sep 2024

**PONE-D-24-32254**Real world data on cervical cancer treatment patterns, healthcare access and resource utilization in the Brazilian public healthcare systemPLOS ONE

Dear Dr. Campuzano,

Thank you for submitting your manuscript to PLOS ONE. After careful consideration, we feel that it has merit but does not fully meet PLOS ONE’s publication criteria as it currently stands. Therefore, we invite you to submit a revised version of the manuscript that addresses the points raised during the review process.

We look forward to receiving your revised manuscript.

Kind regards,

Angelica Espinosa Miranda, M.D., Ph.D.

Academic Editor

PLOS ONE

Journal Requirements:

“I have read the journal's policy and the authors of this manuscript have the following competing interests: TMFC, MACSM, PMB, MACR, MN, LPLG and CPM are employees of Merck Sharp Dohme Farmacêutica Ltda, Sao Paulo, Brazil, who may own stock and/or hold stock options in Merck & Co., Inc., Rahway, NJ, USA. ANR is a scientific medical consultant who was paid by MSD Brazil. PCP and ACS were employees of IQVIA Brazil, which was contracted by MSD Brazil to conduct the study.”

4. Please upload a new copy of Figure 5 as the detail is not clear. Please follow the link for more information: " ext-link-type="uri" xlink:type="simple">https://blogs.plos.org/plos/2019/06/looking-good-tips-for-creating-your-plos-figures-graphics/"
" ext-link-type="uri" xlink:type="simple">https://blogs.plos.org/plos/2019/06/looking-good-tips-for-creating-your-plos-figures-graphics/"

Additional Editor Comments:

The manuscript presents valuable insights based on national Brazilian data. The analysis encompasses a large population, offering findings that are highly relevant to public health policies. However, the paper would benefit from a thorough language review to enhance sentence structure and correct grammatical errors. Additionally, the references should be revised to ensure compliance with the journal's formatting guidelines.

Reviewers' comments:

Reviewer's Responses to Questions

**Comments to the Author**

1. Is the manuscript technically sound, and do the data support the conclusions?

Reviewer #1: Yes

Reviewer #2: Yes

2. Has the statistical analysis been performed appropriately and rigorously? 

Reviewer #1: Yes

Reviewer #2: Yes

3. Have the authors made all data underlying the findings in their manuscript fully available?

Reviewer #1: Yes

Reviewer #2: Yes

4. Is the manuscript presented in an intelligible fashion and written in standard English?

Reviewer #1: Yes

Reviewer #2: Yes

5. Review Comments to the Author

Reviewer #1: Despite being a study of secondary data, the research proposal is interesting and the results, although expected in some way, are surprising. The relationship with geographical issues, especially in the North, surprised me and generated a series of questions. You discussed this part well in the discussion section, but I was encouraged to read more about the geographical disparities, the lower number of cancer diagnoses, the low health indicators and the probable relationship with underreporting. It's a survey of secondary data that paints an alarming picture of cervical cancer in Brazil.

Reviewer #2: In general, the manuscript provides an interesting and useful analysis of a large population, derived from the Brazilian national data, based on healthcare records accumulated over 7 years (2014-2020), and offers evidence to refining the public health policy regarding access to cervical cancer treatment, lines of treatment, and resources utilization. The data supports the conclusions. More information and details are attached.

6. PLOS authors have the option to publish the peer review history of their article (what does this mean?). If published, this will include your full peer review and any attached files.

Reviewer #1: **Yes: **Franco Luís Salume Costa

Reviewer #2: No

---

## [Author Response · Author response to Decision Letter 0]

8 Oct 2024

Dear reviewers and Editor,

First, thank you for your comments and insights. Please find below our answers for each point stated, separated per each reviewer and for the additional comments from the editor:

REVIEWER 1

1. Line 26: “Data from January-2014 until December-2020”. Consider replace ‘until’ with ‘to’. 

Thank you for making this observation. We have updated the manuscript according to your suggestion. 

2. Lines 38-39: “The treatment pattern showed that chemoradiotherapy was the more frequent regimen 39 overall and that the use of chemotherapy increases with staging, as well as the HCRU.”

• Suggestion: The treatment pattern showed that chemoradiotherapy was the most frequent regimen overall and the use of chemotherapy and HCRU increased with staging.

Thank you for bringing this to our attention. The sentence was updated according to your suggestion.

3. Lines 56-57: “has been available for girls from 9 to 14 years since 2014 and 57 for boys from 11 to 14 years since 2017” consider replacing with “has been available for girls aged 9 to 14 e for boys aged 11 to 14” or “has been available for girls from 9 to 14 years old since 2014 and for boys from 11 to 14 years old since 2017 “

Thank you for the correction. It was updated.

4. Line 59: “the most effective exam to detect precancerous abnormalities and prevent CC” – reference? 

• There are recent studies that recognize the HPV DNA or pap smear + HPV DNA as the most effective exam to detect precancerous abnormalities. 

We agree with your observation. The focus of the sentence was to describe the recommendation for cervical cancer (CC) screening according to Brazilian guidelines, which still prioritize the cytology (Pap smear) exam. The current guidelines also recommend the HPV DNA test in conjunction with the Pap smear for evaluating women with colposcopy results showing no abnormal findings or only minor abnormalities. In the paragraph, we highlight that the DNA test for HPV screening should be available in the Brazilian public health system by 2024, according to SECTICS/MS Ordinance No. 3 published in March of this year [Line 64].

5. Line 61: “in case of normal results” – consider replacing with “negative results”. What is a normal result?

We agree that the information was incorrect. According to Brazilian guidelines for CC screening, the correct term is ‘negative results’ and it was adjusted accordingly.

6. Line 63: “Despite national programs for CC cancer prevention” – the word ‘cancer’ is written twice. ‘CC’ (cervical cancer) and ‘carcer’ right after.

Thank you for your insight. The adjustment has been made in line 63 and throughout the document.

7. Line 68: I missed mentioning brachytherapy as a line of treatment. 

Thank you for the insight. We agree the information could be more detailed. It was changed: “For locally advanced/advanced disease (III-IV), the standard is chemoradiotherapy (CCRT), which involves external beam radiation therapy combined platinum-based chemotherapy, followed by brachytherapy.” [Line 69]

8. Line 78: “between regions of the country related to social, economic and health access determinants” – perhaps you should also include ‘geographical’

Thank you for the insight. The term was included in the sentence.

9. Line 79: “decreased efficacy in the detection of disease” be aware of the semantic differences between effectiveness and efficacy.

Thank you for bringing this to our attention. We agree and it was already changed to “Efficiency”: “In addition to the delay for inclusion of new oncology medications in public healthcare system, Brazil presents significant inequalities between regions of the country related to social, economic, geographical and health access determinants, which have been associated with decreased efficiency in the detection of disease, treatment adherence and outcomes” [line 91, in the clean version]

10. Line 80: “To create strategies to reduce the CC burden in Brazil” - It’s more common for native speakers to start a sentence with the gerund as subject than the infinitive form, like “creating strategies to reduce… “

We appreciate the insight and have updated the sentence accordingly.

11. Line 90: “SUS is the universal healthcare system of Brazil, which covers 75% of the population [23]”. Is this a necessary information? Is this a precise information? If the SUS is the ‘universal’ healthcare, how can it cover only 75%? SUS is widely available to all the population. It’s a constitutional right. The fact that 24% of users possess private healthcare coverage doesn’t mean they don’t use the system in some way. 

We agree with your consideration. However, it is a common practice to include this information to clarify that the study does not analyze the entire Brazilian population. This is also mentioned in the limitations section, line 391. The sentence in line 93 has been updated to: “Despite the universal coverage of the Brazilian public healthcare system, approximately 75% rely exclusively on it [23]”.

12. Line 105: “Data from January 2014 until December 2020 was gathered of all patients with the 106 International Classification of Diseases tenth revision for cervical cancer (ICD-10 code C53)”. The reference?

• Consider replace ‘until’ with ‘to’. 

Thank you for your comment. The correction has been made and the reference included.

13. Line 124: Consider replace ‘until’ with ‘to’. You have already said the study’s period. Maybe you could replace the ‘data from … to… ‘ with another expression. 

The expression “From january 2014 to December 2020” was replaced for “during the study period” [line 125].

14. Line 130: “Patient’s stages were considered in the first ICD-10 claim in SIA-AQ and SIA-AR databases (subsets of chemotherapy and radiotherapy from SIA database), which provides stage information.” In the same sentence, you have said that those databases provide stage information twice. Also, you have said that previously in lines 102-103.

Thank you for your comment. The duplicated information was removed, and the paragraph adjusted.

15. Lines 130-132: “Patients with stage inconsistent information (i.e., stage information different from I to IV) at first cervical cancer claim were excluded (Fig 1)” – It would be better written as ‘patients with inconsistent stage information…’

Thank you. The sentence was updated.

16. Line 139: “Additionally, in the staged patient’s distance”. Take the ‘in’ word out.

Thank you. The sentence was updated.

17. Lines 140-142: “(measured as the date of treatment initiation, defined in this study as the time elapsed from the first claim of ICD142 10 C53 to first treatment procedure [CT, RT or surgery – except conization/cone biopsy])”.

• Consider removing the ‘measured as the date of treatment initiation’ and just use the ‘defined in this study…’. They explain the same thing, it’s repetitive. 

Thank you for your insight. It was adjusted.

18. Lines 149-150: “with absolute frequencies and percentages with 95% confidence interval (CI) and percentages”. You’ve written ‘percentages’ twice.

Thank you. The second word was excluded.

19. Lines 153-154: “Thus, the longer the follow-up, lower the PPPM will be, even if there are many procedures claims”. You don’t need the ‘will be’, because you are already using a comparative form.

Thank you for your insight. It was adjusted.

20. Line 157: “The number of LOTs received was summarized.” – I understand this information was already stated in the previous sentence “…and most frequently reported treatments for each line of treatment (LOT) using a Sankey diagram”. 

Thank you for making this observation. We agree it was stated previously. The sentence was excluded.

21. Are the abbreviations ‘pppm’ and ‘hcru’ official or used in other reference? They’re confusing and difficult to associate along the reading. I had to return to the Methods section all the time to understand them. 

The acronym HCRU is widely used to designate Healthcare Resources Utilization, especially in studies conducted using secondary databases or medical records, so we chose to keep it in the text.

As we understand that PPPM may not be so common and may be confused with other acronyms, we chose to replace it throughout the text with "per patient per month". We kept the acronym only in the tables, with its respective legend below.

22. Table 1

• In the title: ‘Description of sociodemographic characteristics of included patients’ instead of ‘(…)patients included’.

It was adjusted. Thank you.

• In ‘Ethnicity N(%)’ the percentage and the absolute numbers doesn’t match with the total included population (206,861). After several calculations, I concluded that the percentages were calculated after the ‘all patients’ minus ‘missing information’ and that’s not clear in the text. Meaning, in this case, the total N is not 206,861, but 178,582. Is that correct? Perhaps you could add an asterisk to clear that data like you did in ‘outpatient visit’ in table 2.

Thank you for bringing this to our attention. We believe the correct percentage should be calculated based on the number of each ethnicity relative to the total population (206,861). The percentages have been adjusted accordingly. 

• You have already said in the Methods section about the age at index date, and in the Results section about the total of staged patients. Is it necessary to write that again in the table?

We agree this information is clear throughout the manuscript and removed it from the table.

23. Table 2:

• Hospitalization rate in days???

Hospitalization was measured as the mean or median number of hospitalizations per patient during the study period. In the Median per patient per month, it is possible to analyze the median number of hospitalizations per patient each month. Then, the Length of Stay (LOS) measures the mean and median number of days hospitalized.

• “Outpatient visit1” – What does that superscripted number refer to?

Thank you for the observation. This number was not related to any information, and it was removed.

• Line 201: “* Includes all patients with stage 1-4, and those non-staged.”. I didn’t find any asterisk in the table. I guess this note refers to the “Outpatient visit1”.

Thank you for the observation. The asterisk “*” belongs to the column “All patients.” The legend for Table 2 clarifies why the sum of the staged patient columns (Stages I-IV = 90,073) does not match the column “All patients” (206,861). The column “All patients” includes all patients with the ICD-10 code C53 in the database, both staged and those with no stage information available. I updated table 1 to have the same asterisk pattern as Table 2.

• Is there any information/data on brachytherapy?

Information on the frequency of brachytherapy use is detailed in the Supplementary Material (Table S3).

• You have already said in the Results section about the total of staged patients. Is it necessary to write that again in the table?

We agree that the information was already mentioned. However, we prefer to keep the legend in these tables to clarify why the sum of the staged patient columns (Stages I-IV = 90,073) does not match the column “All patients” (206,861). The column “All patients” includes all patients with the ICD-10 code C53 in the database, both staged and those with no stage information available.

24. Line 199-200: “Legend: LOS: length of stay; IQR: interquartile range; CI: confidence interval; SD: standard 200 deviation; PPPM: per patient per month.” Those abbreviations were already explained previously. 

We appreciate your suggestion. To facilitate readability, we have opted to include this information in the legend.

25. Line 250 and S2 table: “Patients with stage III and IV have traveled a greater median distance of 7.96 km [IQR 0.0 - 89.16] compared to 4.88 km 252 [IQR 0.0 - 75.16] in the non-advanced stages (S2 Table)” However, in the Supplementary table, ‘4.88 km 252 [IQR 0.0 - 75.16]’ refers to advanced stage IV. You should review the table later. 

Thank you for bringing this to our attention. The table was reviewed, and the sentence updated: “Patients with stage III and IV have traveled a greater median distance of 7.96 km [IQR 0.0 - 89.16] and 4.88 km [IQR 0.0 - 75.16], respectively, compared to the non-advanced stages”

26. S2 Table: The stratification per kilometers is the sum of all the kilometers spent traveling by all the patients on all the visits, correct? I took some time to get that. 

The stratification by kilometers indicates the number of visits to the health center categorized by distance. The percentage reflects the proportion of the total number of visits. This information is detailed in the table legend and has been adjusted for clarity.

27. S3 Table:

• ‘Surgery3 - N (%)’ – What does the superscripted number stand for?

Thank you for bringing this to our attention. The information was missing. It refers to the statement that ‘Patients may undergo multiple surgical procedures,’ which has now been included in the table legend.

• You don’t need the comma in the sentence ‘Cervical cancer treatment, according to disease stage.’

It was excluded. Thank you.

28. Line 259: ‘Lymphadenectomy with pelvic lymphadenectomy’. Try to replace for only ‘pelvic lymphadenectomy’’

Thank you for your insight. It was adjusted.

29. Line 268: “About 90% of stage I patients versus 67% of stage IV patients were submitted to RT (Error! Reference source not found.).” I don’t think there’s any reference in there.

Thank you for your comment. This reference refers to ‘Figure 4B’. It was linked automatically by Word’s ‘Cross-reference’ feature, which likely contained an error. It has now been updated according to PLOS ONE guidelines.

30. Line 295: Avoid using the first-person plural. Suggestion: ‘so far’.

Thank you for the insight. It was adjusted.

31. Line 297: ‘Although there were improvements in the prevention of cervical cancer in Brazil’. The prevention policies continue to improve, perhaps you should write as: “Although there have been improvements… “

Thank you for the insight. It was adjusted.

32. Line 299: “Like our results” –Avoid using the first-person plural. Suggestion: ‘Similar to these findings, other Brazilian studies’

Thank you for the insight. The sentence was changed: “Similar to the findings of this study, other Brazilian studies reported a median age of 50 years in CC patients…”

33. Line 300: mean or median?

Thank you for your comment. The information refers to the mean age found in other Brazilian studies and has been adjusted accordingly

34. Line 301: ‘were’ not ‘are’

It was adjusted. Thank you.

35. Line 324: Replace ‘in most wealthy’ with ‘in wealthier.’

It was adjusted. Thank you.

36. Line 324: “what was mentioned about health access and population’s well-being” – mentioned where? What about health access and population’s well-being?

Thank you for your observation. We were referring to the previous paragraphs. We agree it was not evident and it was adjusted: “The higher concentration of patients in wealthier Brazilian regions reinforces the relationship between greater access to healthcare and earlier disease diagnosis, which can result in better outcomes.”

37. Line 325-326: “as with higher income per-capita” – ‘per-capita income’

It was adjusted. Thank you.

38. Lines 325-327: “since these regions are characterized as with higher income per-capita, educational levels, concentration of health services, access to reference medical units and higher health resources per-capita in both private and public sectors in these regions” – ‘these regions’ is mentioned twice in the same sentence. Perhaps you should consider turn this sentence into two different ones, so it doesn’t get confused.

Thank you for the insight. We agree the sentence was loo long, and it has been adjusted: “The higher concentration of patients in wealthier Brazilian regions reinforces the relationship between greater access to healthcare and earlier disease diagnosis, which can result in better outcomes [32]. These areas are characterized by higher per-capita income, elevated educational levels, concentration of health services, access to reference medical units, and hig

---

## [Editor Report · Decision Letter 1]

14 Oct 2024

Real world data on cervical cancer treatment patterns, healthcare access and resource utilization in the Brazilian public healthcare system

PONE-D-24-32254R1

Dear Dr. Campuzano,

We’re pleased to inform you that your manuscript has been judged scientifically suitable for publication and will be formally accepted for publication once it meets all outstanding technical requirements.

Kind regards,

Angelica Espinosa Miranda, M.D., Ph.D.

Academic Editor

PLOS ONE
---

## [Editor Report · Acceptance letter]

20 Oct 2024

PONE-D-24-32254R1 

PLOS ONE

Dear Dr. Campuzano, 

I'm pleased to inform you that your manuscript has been deemed suitable for publication in PLOS ONE. Congratulations! Your manuscript is now being handed over to our production team.

Kind regards, 

on behalf of

Dr. Angelica Espinosa Miranda 

Academic Editor

PLOS ONE